# Neurofilament-Light, a Promising Biomarker: Analytical, Metrological and Clinical Challenges

**DOI:** 10.3390/ijms241411624

**Published:** 2023-07-19

**Authors:** Salomé Coppens, Sylvain Lehmann, Christopher Hopley, Christophe Hirtz

**Affiliations:** 1National Measurement Laboratory, LGC Limited, Teddington TW11 0LY, UK; salome.coppens@lgcgroup.com (S.C.); christopher.hopley@lgcgroup.com (C.H.); 2Univ. Montpellier, IRMB-PPC, INM, CHU Montpellier, INSERM CNRS, 34295 Montpellier, France; s-lehmann@chu-montpellier.fr

**Keywords:** neurofilament-light, neurodegenerative disease, biomarker, standardization, implementation

## Abstract

Neurofilament-light chain (Nf-L) is a non-specific early-stage biomarker widely studied in the context of neurodegenerative diseases (NDD) and traumatic brain injuries (TBI), which can be measured in biofluids after axonal damage. Originally measured by enzyme-linked immunosorbent assay (ELISA) in cerebrospinal fluid (CSF), Nf-L can now be quantified in blood with the emergence of ultrasensitive assays. However, to ensure successful clinical implementation, reliable clinical thresholds and reference measurement procedures (RMP) should be developed. This includes establishing and distributing certified reference materials (CRM). As a result of the complexity of Nf-L and the number of circulating forms, a clear definition of what is measured when immunoassays are used is also critical to achieving standardization to ensure the long-term success of those assays. The use of powerful tools such as mass spectrometry for developing RMP and defining the measurand is ongoing. Here, we summarize the current methods in use for quantification of Nf-L in biofluid showing potential for clinical implementation. The progress and challenges in developing RMP and defining the measurand for Nf-L standardization of diagnostic tests are addressed. Finally, we discuss the impact of pathophysiological factors on Nf-L levels and the establishment of a clinical cut-off.

## 1. Introduction

Neurofilament-light chain (Nf-L) is a reliable biomarker in the context of neurodegenerative diseases (NDD) and traumatic brain injuries (TBI). This protein is found principally in large-caliber myelinated axons and is part of a complex called neurofilament (Nf). Nf-L is associated with other intermediate filaments (IF) subunits: neurofilament-heavy chain (Nf-H), medium chain (Nf-M) and alpha-internexin in the central nervous system (CNS) or peripherin in the peripheral nervous system (PNS) [1,2]. They are in general composed of three domains: the head, the rod and the tail. Parts of the rod domain are relatively conserved among the IF as it is essential to enhance the polymerization of the Nf. Once assembled, Nfs are located mainly in the axons and organized as a network where they are essential to maintain the axon caliber, to ensure the radial growth and the transmission of electrical impulses [1,2]. Under physiological conditions, low amounts of Nf-L are released during brain development, maturation and aging. However, when axonal damages or neuronal degeneration occur, it is released in larger quantities into the brain interstitial fluid, cerebrospinal fluid (CSF) and blood. It has been recorded that Nf-L can be monitored in several pathological conditions such as neurodegenerative disease and dementias, but also after a stroke or a traumatic brain injury (TBI). It has therefore been identified as a non-specific marker for axonal damage and neurodegeneration, and several studies have been conducted to measure it in biofluids. Currently, the quantification techniques performed to measure this biomarker in biofluid are based on immunoassays. The first enzyme-linked immunosorbent assay (ELISA) measuring Nf-L was developed in 1996 by Rosengren L. et al. [3]. Since then, other ELISA and more sensitive assays such as chemiluminescent immunoassay (CLIA), electrochemiluminescent assay (ECL) or single molecule array (Simoa^®^) have been developed with the aim of measuring extremely low concentrations in complex matrices like blood [4]. While the performance of immunoassays quantifying Nf-L is good, they have not been implemented in clinical practice yet as several challenges remain, including standardization and the identification of reliable clinical thresholds.

Here, we will summarize the current context of Nf-L, highlighting the different associated diseases, biofluids and diagnostic purposes as well as the current quantification techniques showing great potential for clinical implementation. Then, we will focus on the remaining challenges around this biomarker. We will first describe the measurement variability and the pre-analytical component affecting Nf-L levels in biofluids. Furthermore, we will focus on the metrological challenges such as developing RMP and CRM and on characterizing the measurand, and how mass spectrometry could help to address them. These considerations are critical to achieving standardization of the assays and the development of assay calibrants in order to ensure long-term success. Finally, we will also describe the remaining clinical challenges, including the different pathophysiological factors affecting Nf-L measurements and the need for defining reliable clinical thresholds prior to implementation in clinical routines.

## 2. Clinical Applications

Nf-L is a well-established biomarker in the clinic for assessing axonal damage. However, it is not specific to a disease as its level increases in several pathologies such as multiple sclerosis (MS), amyotrophic lateral sclerosis (ALS), Huntington’s disease (HD), dementias including Alzheimer’s disease (AD), fronto-temporal dementia (FTD) and other diseases [4,5,6,7,8,9]. Nf-L levels are also higher in days following a stroke or after a TBI [4,5]. Therefore, if this biomarker is to be used for diagnostic purposes, it should be complemented with disease-specific biomarkers. 

Nevertheless, in some instances, Nf-L can differentiate one disease from another and help make differential diagnoses. This is the case with Parkinson’s disease (PD) and FTD. Indeed, a higher level of Nf-L has been shown to correlate with atypical Parkinsonian disorder than with PD. A study demonstrated a good diagnostic accuracy to differentiate both with a receiver operating characteristic (ROC) curve between 0.81 and 0.91 [10]. Similarly, FTD can be discriminated from primary psychiatric disorders [11]. Behavioral FTD (bvFTD) is a variant of the disease that often presents overlapping symptoms with psychiatric disorders. For those, a differential diagnosis is needed. Several studies have demonstrated that CSF and serum Nf-L is higher in patients suffering from bvFTD than in those with psychiatric disorders, which did not show a significant difference with controls [12,13].

Additionally, its utility has been highlighted not only for diagnostic but also for prognostic purposes and for evaluation of treatment efficiency. Indeed, several studies have shown that an increase in Nf-L levels correlates with disease evolution. For instance, in patients suffering from MS, a rise in Nf-L concentration can predict relapses. It is also the case for AD patients. Indeed, different values in Nf-L measurements were observed between cognitively impaired and AD dementia patients [7,9]. In addition, increased Nf-L levels were associated with cognitive decline in the AD continuum and can predict disease progression. It also correlates highly with imaging and cognitive assessment results [14,15]. 

Moreover, as with many other NDD biomarkers, Nf-L levels increase before any clinical manifestations. A rise of Nf-L can be seen as early as 10 years in sporadic AD before any clinical symptoms appear [16]. In familial AD, it could even be 22 years before as demonstrated by Quiroz et al. [17] in a study comparing presenilin 1 E280A mutation carriers with non-carriers. In addition, the yearly increase in Nf-L was much more observable in mutation carriers. The average annual change was 15.26 pg/mL for mutation carriers vs. 1.99 pg/mL for non-carriers. 

Finally, monitoring Nf-L in clinical trials showed an enormous potential to determine treatment efficiency. Raket et al. [18] supported the use of Nf-L in clinical trials for monitoring the cognitive decline occurring in the short term. Furthermore, in several studies, the effect of treatments for multiple sclerosis (MS) was evaluated by monitoring Nf-L levels in the blood. Kuhle et al. [19,20] described the effect of fingolimod and alemtuzumab in MS patients using serum Nf-L measurement for monitoring the efficacy of treatment. In the first study, they showed that Nf-L was sensitive to the treatment and indicated a decrease of 36% after 6 months in patients treated with fingolimod in two trials compared to a placebo [19]. In the second study, they monitored Nf-L levels during treatment with alemtuzumab and 7 years post-treatment. They showed that Nf-L levels decreased at the beginning of the treatment and were stable over the 7 years of follow-up [20].

## 3. Analytical Challenges

Nf-L quantitation in body fluids is performed currently by immunoassays using various technologies (Table 1) but some challenges in measurement remain.

Pre-analytical variations including the type of matrix or the sampling procedure and sample storage are important considerations, which will be discussed below in greater detail.

### 3.1. Nf-L Measurements in Body Fluids

Initially measured using semi-quantitative methods such as immunoblotting, Nf-L concentrations are now measurable in blood and its derivatives owing to the advent of ultrasensitive assays. Nf-Lin human CSF was first measured with ELISA developed by Rosengren et al. in 1996 [3]. Subsequently, more assays have followed such as the commercially available UmanDiagnostic assay for measurement in CSF samples but more recently it has become possible to measure Nf-L in serum with a limit of quantification (LOQ) of 0.8 pg/mL. Other assays developed, such as those described by Gaetani et al. [21] and Das et al. [22], rely on newer ELISA methods with a LOQ of 78 pg/mL [21] and 100 pg/mL [22] in CSF, respectively. Finally, due to the development of ultrasensitive assays such as microfluidic platforms, e.g., Ella (ProteinSimple), ECL assays including Meso Scale Discovery (MSD) from Meso Scale Diagnostic or Attelica Solutions (Siemens), and the Simoa (Quanterix), measurement of Nf-L in blood and its derivatives have become easily achievable with LOQs in the low pg/mL range (Figure 1). Other manufacturers are developing automated assays for the quantification of Nf-L, such as Roche on the Cobas platform and Fujirebio on the Lumipulse. The use of non- or minimally invasive sampling methods such as blood collection is now preferred compared with lumbar puncture. The transition from CSF to blood has greatly improved patient well-being and allowed more samples to be obtained more frequently. Indeed, measurement in serum or plasma permits more longitudinal studies thereby improving the utilization of Nf-L for monitoring disease activities and treatment effects [23]. A reasonably good correlation between CSF and blood Nf-L levels has been established in various studies. For instance, Kuhle et al. [24] demonstrated a good correlation with an r^2^ of 0.88 between CSF and serum Nf-L measurements using the Simoa platform and 0.78 with the MSD. Similar results were obtained for Nf-L in CSF and blood samples from two cohorts using the Simoa platform [10]. 

Despite the emergence of sensitive techniques and a better understanding of Nf-L as a biomarker, some challenges remain to be addressed before fully implementing Nf-L in clinical practice. First of all, there is a lack of standardization for this marker. In the meta-analysis published by Forgrave et al. [25], whilst results correlated large differences in absolute values for Nf-L quantification were observed across the numerous studies reviewed. Therefore, standardization of Nf-L measurement would be valuable for inter-laboratories studies but also for establishing reliable cut-off values for this biomarker. Additionally, identifying which analytical and clinical factors impact Nf-L levels is also important to define those thresholds. 

### 3.2. Analytical Variability

#### 3.2.1. Type of Sample and Sampling Effects

Nf-L values may vary depending on the type of body fluid in which it is measured. Nf-L in CSF is often more concentrated than in blood due to its proximity to the brain. This difference is also present between plasma Nf-L and serum Nf-L. For plasma, the type of collection tube used for sampling has an impact on the total Nf-L levels measured. For example, Ashton et al. [26] measured plasma Nf-L collected in tubes with different additives: lithium-heparin (LiHep), ethylenediaminetetraacetic acid (EDTA), citrate or NF-L in serum. Nf-L values were slightly different between the four types of collection. Serum and LiHep-plasma showed the highest concentrations as compared to EDTA-plasma and citrate-plasma. EDTA-plasma and serum are the most commonly used, nevertheless is not clear yet which one is the most suitable, the selection for one or the other being specific to the disease or clinical center: for some neurodegenerative diseases such as ALS or MS, the serum is preferred (328 publications on PubMed vs. 104 in plasma), whereas, for dementias, plasma is more often used (382 publications vs. 209 in serum). Furthermore, study comparisons are sometimes difficult due to the preferential use of either plasma or serum. Some researchers have therefore established a correlation between EDTA-plasma and serum [27].

#### 3.2.2. Sample Handling and Storage

Sample treatment is also a source of variability as storage temperature, delayed freezing and multiple freeze–thawing cycles can alter the sample and affect the measurement. Nonetheless, good stability for this marker can be achieved after 3–4 freeze–thawing cycles and where freezing is delayed to 24 h at 2–8 °C or room temperature [26,28,29].

## 4. Metrological Challenges

To ensure clinical implementation, an RMP and CRM should be developed to allow standardization and traceability of results over time and space. Efforts are ongoing to standardize the measurement of Nf-L through various initiatives, e.g., the CSF working group of the International Federation of Clinical Chemistry and Laboratory Medicine (IFCC). Powerful techniques such as mass spectrometry have been employed to develop both CRMs and RMP but also to define the measurand.

### 4.1. Developing Certified Reference Materials and Reference Measurement Procedures

Achieving traceability is essential to reduce the inter-laboratory and inter-method variability stemming from different methods and/or choices of calibrants. The ultimate aims are to ensure patient safety when diagnosis is established and to underpin the regulatory approval of the assays [30]. For the latter, this is especially true as new regulations on in vitro diagnostics (IVD) will be implemented by 2027 [31], requiring documented RMP and CRMs for IVD medical devices to ensure traceability.

By definition, metrological traceability is the property of a measurement result associated with a reference [32]. This is ensured and documented through a reference measurement system with an unbroken traceability chain as presented in Figure 2 and its principles are described in ISO 17511:2020 [33]. A reference measurement system includes two main components: CRMs or calibrators and reference measurement procedures.

#### 4.1.1. Certified Reference Materials

CRMs are stable and sufficiently homogeneous materials which are fit for their intended use. They should reflect the properties and the quantities of the analyte in clinical samples. For these materials, providers must state a value that is assigned by a metrologically valid method and associated with a defined uncertainty value [32]. In general, when a CRM is available, it is used as a calibrator in the development of RMP. Currently, there is no available CRM for Nf-L. This is due to its structural complexity and the presence of multiple modifications, isoforms and/or fragments, especially in biofluids [34]. This is the case for most proteins in the field of NDD except for amyloid-beta 1-42 (Aβ 1-42) for which three matrix-based CRMs have been developed (ERM^®^-DA480/IFCC, ERM^®^-DA481/IFCC and ERM^®^-DA482/IFCC). Their values were assigned through the development of an RMP and they have been used to successfully recalibrate three commercial tests [35]. However, when no CRM is available, other sources of materials must be used to develop RMP such as commercially available standards. Those need to be characterized prior to their utilization in the development of the reference method. Nevertheless, investigations are ongoing to find the best way to develop a matrix-based CRM for Nf-L in blood. In a recent paper, several CRM candidates have been assessed for commutability [36]. Commutability is the closeness of agreement between the relation among the measurement results for a stated quantity [32]. This property is essential for a CRM. Here, the authors analyzed candidate materials composed of plasma or serum only or spiked with a known quantity of CSF or recombinant Nf-L in parallel with clinical samples (paired plasma and serum) on multiple immunoassay platforms. Good commutability was observed for plasma and serum spiked with CSF containing Nf-L unlike those spiked with recombinant protein. This study demonstrates the feasibility of developing such a CRM for Nf-L in blood but, as indicated by the authors, the matrix (plasma or serum) used must be chosen in accordance with clinical need. Furthermore, in order to assign a value to this CRM, the concentration of CSF used should be certified and for this, an RMP is necessary [36].

#### 4.1.2. Reference Measurement Procedure

RMPs enable assessment of the measurement trueness of measured quantity values obtained from other measurement procedures which are measuring quantities of the same kind for calibration or characterization of reference materials [32]. Those methods are developed by National Measurement Institutes (NMIs) and/or Designated Institutes (DIs) and are often based on liquid chromatography coupled to mass spectrometry (LC-MS) approaches. It includes the selection and characterization of a primary calibrator, the development of the quantification method and its validation according to the criteria in ISO 17511:2020 [33]. 

In the field of the NDD biomarker, two were successfully developed, approved and published for Aβ 1-42 on the JCTLM (Joint Committee for Traceability in Laboratory Medicine) databases as C12RMP1 [37] and C11RMP9 [38]. However, no RMP for Nf-L has been approved yet although one is currently under development [39]. Developing such methods for proteins in general is challenging and not as easy as for small molecules. This is due to the protein complexity (e.g., isoforms, post-translational modifications, fragments and multimers) and the lack of information about the measurand [34]. Moreover, achieving the concentration measured by immunoassays, especially in blood, since ultrasensitive assays have been developed is also difficult to reach by mass spectrometry. Solutions and alternatives should be found to standardize all of these assays. An alternative proposed by Andreasson et al. [36] would be to gravimetrically spike quantities of CSF quantified by a validated CSF RMP to establish a blood RMP for Nf-L.

### 4.2. Defining the Measurand

Several studies, in CSF and in blood, have demonstrated that circulating forms of Nf-L are different from the full-length protein localized in neurons [16]. Some approaches have already been developed to better understand which fragments are present and in which fluid. 

In 2017, Brureau et al. [40] observed that Nf-L immunoprecipitated from the CSF of p25 mouse and run on SDS-PAGE was not the full-length protein but a fragment of it. The bands were cut out and tryptic-digested prior to analysis by LC-MS. One fragment at 10 kDa was identified as Nf-L based on specific peptides identified by the mass spectrometry analysis [40]. 

Another publication presented an approach based on immunoprecipitation-mass spectrometry (IP-MS) for the identification of Nf-L fragments [41]. Several monoclonal antibodies with known epitopes located on different domains of the Nf-L sequence were used, which were, respectively, localized to coil 1 and coil 2 of the rod domain and on the tail domain of the protein. The head domain was not covered. Sequential IPs were performed on CSF prior to tryptic digestion and LC-MS analysis. Using this method, at least three major fragments of Nf-L were identified, showing that several species must be present in CSF [41]. 

Finally, fragments of Nf-L were also identified in blood in two other studies [42,43]. One evaluated the use of dry-blood spots (DBS) for Nf-L quantification in ALS patients and identified several fragments after Western blotting. A 22 kDa fragment was identified in the plasma of healthy controls and ALS patients as well as a 52 kDa in DBS eluates [43]. In addition, another study of Nfs in blood protein aggregates demonstrated the presence of a 31 kDa fragment by Western blotting [42]. 

These studies suggest the presence of matrix-specific fragments which are highly dependent on the proteases present in the different biofluids. Thus, the characterization of the different proteoforms measured by immunoassays in biological fluids is also essential for the standardization and development of assay calibrants and CRMs [44]. As shown by Budelier et al. [41], mass spectrometry is capable of identifying and quantifying the different isoforms. Unlike others, this technique is particularly powerful for the characterization of isoforms, proteoforms and fragments and can offer a real added value as compared to immunoassays by detecting and quantifying the different Nf-L species present in biofluids [40,41]. 

## 5. Clinical Challenges

Identifying reliable clinical thresholds would be invaluable from the clinical point-of-view to better stratify the different patient groups. It is known that several pathophysiological conditions result in different Nf-L concentrations in body fluids and a better understanding of the clinical background is essential prior to the establishment of cut-off values. 

### 5.1. Factors Influencing Nf-L Levels

Nf-L levels are influenced by several factors like gender, age, comorbidities, or even body mass index (BMI). Here we summarize their effects on Nf-L levels.

#### 5.1.1. Gender

Numerous studies have tried to better understand the link between Nf-L levels and the sex of the patients. However, no correlation has been established yet. Some have reported higher levels of Nf-L among men than women. It was suggested that the higher proportion of white matter in men’s brains contributes to these results [4,45]. Other studies have highlighted higher levels in females with ALS due to the higher severity of the disease among women [4]. 

#### 5.1.2. Age

It is established that Nf-L concentration correlates and varies with the age in healthy controls and patients. Indeed, during the aging process, it was shown that Nf-L levels were rising. This is because, during the brain aging process, neuronal loss occurs and the blood–brain barrier becomes more permeable [4,45,46]. In 2019, Bridel et al. [47] showed a 3.3% increase in CSF Nf-L per annum in healthy controls. In another study, Khalil and co-workers [46] described the percentage of yearly increase to be 2.2% in blood Nf-L, but 4.3% in the >60-year-old population, although in this population, higher variability in Nf-L levels was observed. This might be due to the effect of comorbidities which are more prevalent in the elderly population [4,46]. 

#### 5.1.3. Body Mass Index

Several studies have reported a strong inverse correlation between BMI and Nf-L levels. Individuals with higher BMI and therefore larger blood volume have in general a lower Nf-L concentration when measured in the blood due to dilution effects [48,49]. Interestingly, CSF Nf-L is not affected by this phenomenon [48]. Moreover, this effect is independent of other factors, such as age, kidney function or cardiovascular comorbidities [49]. Therefore, this parameter needs to be considered by clinicians when blood measurements are performed. 

#### 5.1.4. Renal Function

It was shown that plasma and serum Nf-L are positively correlated with the creatinine concentration and negatively correlated with the estimated glomerular filtration rate as described by Akamine et al. [50] and Polymeris et al. [49]. When kidney function is altered, all waste products may not be eliminated by the organs, decreasing the clearance of some markers. Higher levels of Nf-L are observed in individuals of any age with kidney failure independently of other factors or comorbidities [49,50,51]. Therefore Nf-L level needs to be adjusted in correlation with renal function.

#### 5.1.5. Comorbidities

Comorbidities such as cardiovascular risk or diabetes also have an impact on Nf-L concentrations, especially in the blood. Patients with a history of stroke, cardiovascular lesion or diabetes show higher Nf-L concentrations [52]. 

### 5.2. Establishing Thresholds

As described above, Nf-L is a dynamic biomarker. Its concentration varies significantly depending on physiological and pathological factors. Therefore, defining reliable predictive cut-off values is paramount for better diagnosis. Cut-offs are normally defined in healthy controls; however as Nf-L also increases with age, it must be defined taking patient age into account. 

The first study was published in 2020 by Hviid and co-workers [53] and determined age-stratified intervals based on 342 healthy subjects from 18- to 87-year-olds. Nf-L quantifications were performed in serum using the Simoa assay and reference values were calculated for every 10 years interval. The upper limits were established as follows: 7.4 ng/L for the 20-year-olds, 9.9 ng/L for 30-year-olds, 13.1 ng/L for 40-year-olds, 17.5 ng/L for 50-year-olds, 23.3 ng/L for 60-year-olds, 30.9 ng/L for 70-year-olds, 41.3 ng/L for 80-year-olds and 54.9 ng/L for 90-year-olds. In addition, they estimated a 3% increase per year in Nf-L levels. However, as this study was performed on a very small group of subjects and from the same ethnic group, larger future studies including different populations are needed.

In a recent publication, Simrén et al. [54] also reported thresholds determined for Nf-L values according to age groups. About 1700 blood samples, mainly plasma, coming from 8 cohorts of patients from 5 to 90 years old with no neurodegenerative disorder history, were quantified using a Simoa platform. Cut-off values were established per age group as follows: 7 pg/mL for 5–17-year-olds, 10 pg/mL for 18–50-year-olds, 15 pg/mL for 51–60-year-olds, 20 pg/mL for 61–70-year-olds, and 35 pg/mL for >70-year-olds. These cut-off recommendations are slightly lower than those of the previous study [53], but it can be explained by the difficulties to find elderly subjects with no comorbidities as stressed by both publications [53,54]. In contrast to the study carried out by Hviid and colleagues, this second study benefited from a larger sample size, enabling more populations to be included. However, as these cohorts were mainly of European origin, it would be interesting to include other populations in future studies to harmonize these thresholds globally.

Currently, some online tools are available for physicians, allowing them to take a decision based on cut-offs. For example, Amsterdam UMC developed an interface (https://mybiomarkers.shinyapps.io/Neurofilament/, accessed on 26 June 2023) in which the measured values of Nf-L and patient age can be visualized and interpreted among several reference group values using different percentile intervals and for different types of neurodegenerative diseases.

## 6. Conclusions

This review describes the current limitations of bioassays for Nf-L and their implementation in clinical practice. The major untackled challenges relate to the variability of results and to the numerous clinical and analytical factors affecting the measurement of this biomarker. 

Despite the use of sensitive and robust instruments for the quantification of Nf-L in biological fluids, the obtained values vary considerably which can lead to misinterpretation. Standardization of Nf-L measurement would be invaluable to ensure the harmonization of values globally. The development of CRM and RMP, not only in CSF but also in blood, is crucial in the near future to support the diagnostic use of Nf-L in clinics. Moreover, the factors influencing Nf-L levels outlined in this review are important and must be considered carefully. Besides the urgent need for measurement standardization, the establishment of cut-off values will be key for accurate use of Nf-L for diagnosis, prognosis and patient follow-up.

## Figures and Tables

**Figure 1 ijms-24-11624-f001:**
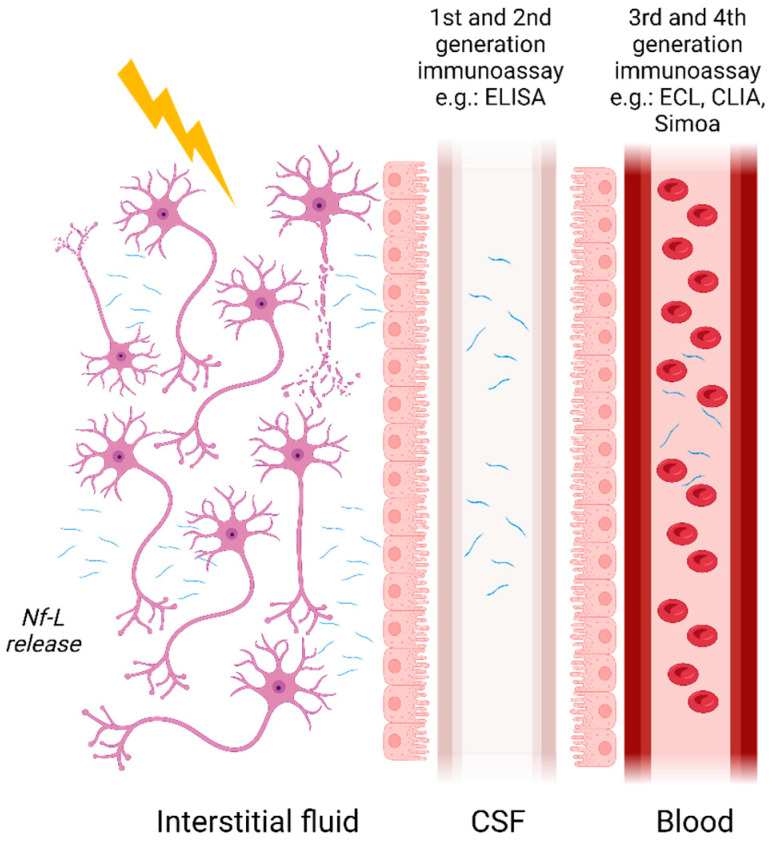
The release of Nf-L in biofluids and the available immunoassay methods for its measurement in different matrices. Nf-L enters from the interstitial fluid into CSF and crosses the blood–CSF barrier making its way to the bloodstream.

**Figure 2 ijms-24-11624-f002:**
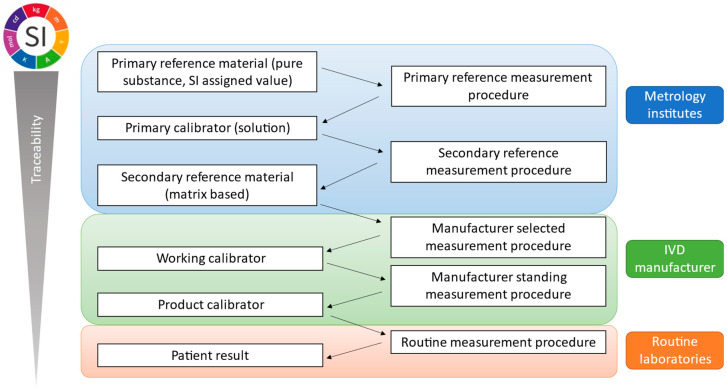
Metrological traceability chain outlined by ISO 17511:2020 indicating elements of the traceability chain across metrology institutes, IVD manufacturers and routine laboratories.

**Table 1 ijms-24-11624-t001:** Available immunoassay technologies for Nf-L measurement according to the type of assay, provider and matrix in which it is applied.

Assay Type	Provider/Publication	Matrix	Assay Range	LOD ^1^	LOQ ^2^
ELISA	Uman Diagnostic	CSF	50–5000 pg/mL	33 pg/mL	81 pg/mL
	Uman Diagnostic	Plasma/Serum	0.5–40 pg/mL	0.4 pg/mL	0.8 pg/mL
	Gaetani et al. [21] Das et al. [22]	CSF CSF	39–5000 pg/mL 50–40,000 pg/mL	- -	78 pg/mL 100 pg/mL
ECL/CLIA	Meso Scale Discovery	Plasma/Serum	5.5–50,000 pg/mL	5.5 pg/mL	-
	Siemens Healthineers	Plasma/Serum	1–646 pg/mL	1.49 pg/mL	1.85 pg/mL
Microfluidic platform (Ella)	Protein Simple	Plasma/Serum and CSF	2.7–10,290 pg/mL	2.7 pg/mL	-
Simoa	Quanterix	Plasma/Serum	0.5–500 pg/mL	0.038 pg/mL	0.174 pg/mL

^1^ Limit of detection; ^2^ Limit of quantification.

## Data Availability

Not applicable.

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
