# Peer review of "Neurofilament-Light, a Promising Biomarker: Analytical, Metrological and Clinical Challenges"

_ijms, 2023, doi:10.3390/ijms241411624_

Round 1
Reviewer 1 Report
The review manuscript by Coppens and co-workers provides a comprehensive review of the current methods used for quantification of Neurofilament-light (Nf-L), a promising biomarker. The authors summarize the progress and current limitations in the field, highlighting the major challenges associated with variability and related analytical factors. They emphasize the importance of developing reliable clinical thresholds and reference measurement procedures (RMPs). In Section 4, the manuscript points out the urgent need for standardization of Nf-L measurement, which is invaluable for advancing research in this area. The various factors that influence Nf-L levels in a thorough manner were also discussed. Overall, this manuscript is well-written, well-organized, and supported by appropriate and adequate references. I believe this review will make a great contribution to the field of Neurofilament-light. Therefore, I agree to accept this manuscript for publication in IJMS after minor revisions.
Please find the specific points for revision below:
In line 50, ".... in complex matrices like blood," it would be beneficial to include a reference to support this statement.
Please review the format of each reference for consistency. Some journal names are provided in full, while others are abbreviated. For example, reference 1 (Journal of Cell Science) and reference 3 (J. Neurochem.). Additionally, please ensure that the journal names are followed by the appropriate punctuation, such as "J. Neurochem." instead of "J Neurochem".
Author Response
Thank you for your comments. Please see the answers to each comment in the attached file.

Reviewer 2 Report
It is a good review about a biomarker of neurodegenerative diseases (NDD) and traumatic brain injuries (TBI), the neurofilament-light chaiin (Nf-L).
I have few observations, in general I like the way you develop the work. You may have mentioned, in which populations the work was developed, in the case of the thresholds according to age groups for populations studied by Hviid et. al. and Simrén et. al. It's good to discuss that.
And in table 1. the authors must indicate what is the LOD and LOQ. I know there is in the text, but to make it easier for the readers must be down below the table.
Author Response

(The authors gave the same response as above.)
